# Bounded Suboptimal Weight-Constrained Shortest-Path Search via Efficient Representation of Paths

## Abstract

In the Weight-Constrained Shortest-Path (WCSP) problem, given a graph in which each edge is annotated with a cost and a weight, a start state, and a goal state, the task is to compute a minimum-cost path from the start state to the goal state with weight no larger than a specified weight limit. While most existing works have focused on solving the WCSP problem optimally, many real-world situations admit a trade-off between efficiency and a suboptimality bound for the path cost. In this paper, we propose a novel bounded suboptimal WCSP algorithm called WC-A*pex that is built on a state-of-the-art approximate bi-objective search algorithm called A*pex. WC-A*pex uses an efficient, albeit approximate, representation of paths with similar costs and weights to compute a $(1 + \varepsilon)$-suboptimal path, for a user-specified $\varepsilon$. During search, WC-A*pex avoids storing all paths explicitly and thereby reduces the search effort while still retaining its $(1+\varepsilon)$-suboptimality property. On benchmark road networks, our experimental results show that WC-A*pex with $\varepsilon = 0.01$ (i.e., with $1\%$ suboptimality) achieves up to an order-of-magnitude speedup over WC-A*, a state-of-the-art WCSP algorithm, and its bounded suboptimal variant.

## 1 Introduction and Related Work

In the Weight-Constrained Shortest-Path (WCSP) problem, given a graph in which each edge is annotated with a cost and a weight, a start state, and a goal state, the task is to compute a minimum-cost path from the start state to the goal state with weight no larger than a specified weight limit. The WCSP problem appears in many real-world applications. In an electric vehicle domain, the graph represents a road network and each edge is annotated with a cost corresponding to driving time and a weight corresponding to battery consumption (Baum et al. 2015). A desired route minimizes the driving time without depleting the battery. In a bicycling domain, the graph represents a road network and each edge is annotated with a cost corresponding to bicycling time and a weight corresponding to climbing altitude gain (Storandt 2012). A desired route minimizes the bicycling time with a user-specified limit on total climbing altitude gain.

Combinatorially, the WCSP problem also appears as a subproblem in the context of column generation methods used for solving other problems, such as the shift scheduling problem (Cabrera et al. 2020) and the virtual network embedding problem (Rost 2019). Although many path-finding problems are tractable, the WCSP problem is NP-hard to solve optimally, i.e., it is NP-hard to compute the minimum-cost path within the weight limit (Handler and Zang 1980; Lorenz and Raz 2001).

The WCSP problem is similar to the Bi-Objective Shortest-Path (BOSP) problem, where each edge is annotated with two costs. Although the task in the BOSP problem (Sec. 3) is different from the task in the WCSP problem, several techniques of BOSP algorithms can be carried over to the WCSP domain by treating the weight as the second cost while being cognizant of the weight limit. In fact, WC-A* (Ahmadi et al. 2022a) is a state-of-the-art WCSP algorithm that draws inspiration from BOSP algorithms. WC-A* and its bi-directional variant, WC-BA* (Ahmadi et al. 2022b), have been shown to outperform previous state-of-the-art algorithms, Bi-pulse (Cabrera et al. 2020) and RC-BDA* (Thomas, Calogiuri, and Hewitt 2019), by up to two orders of magnitude on road networks (Ahmadi et al. 2022a,b).

While the algorithms mentioned above focus on solving the WCSP problem optimally, many real-world situations admit—or even encourage—a trade-off between efficiency and a suboptimality bound for the path cost. A bounded suboptimal WCSP algorithm computes a $(1 + \varepsilon)$-suboptimal path, for a user-specified $\varepsilon$. A $(1 + \varepsilon)$-suboptimal path has a cost within $(1 + \varepsilon)$ times the minimum path cost and a weight that is no larger than the weight limit.

There are only a few existing works on solving the WCSP problem bounded-suboptimally. In the paper of Bi-pulse (Cabrera et al. 2020), the authors provided a straightforward method to convert an optimal WCSP algorithm to a bounded suboptimal one, that is, to terminate the search immediately after the cost of the incumbent solution (i.e., the best solution that an algorithm has found thus far) is proven to be within the given suboptimality bound. Other works on bounded suboptimal WCSP algorithms (Lorenz and Raz 2001; Ergun, Sinha, and Zhang 2002) are typically based on fully polynomial-time approximation schemes, whose runtime is polynomial in the size of the graph and $1/\varepsilon$. Unfortunately, these algorithms are still impractical for large graphs, such as road networks, that often have millions of states.

There are many existing works on bounded suboptimal search algorithms for (unconstrained) shortest-path problems. These algorithms include WA* (Pohl 1970), fo-

cal search (Pearl and Kim 1982), and explicit estimation search (Thayer and Ruml 2011). While speeding up the search via allowing suboptimality is intuitive, it is unclear how to efficiently do so for the WCSP problem.

In this paper, we propose a novel bounded suboptimal WCSP algorithm called WC-A\*pex. WC-A\*pex takes a WCSP instance and a user-specified $\varepsilon \geq 0$ as input and computes a $(1 + \varepsilon)$-suboptimal path. WC-A\*pex imports techniques from A\*pex (Zhang et al. 2022b), a state-of-the-art approximate BOSP algorithm. Unlike other WCSP algorithms, it uses a clever data structure that merges paths with similar costs and weights (instead of storing them explicitly) during the course of its search. Since paths correspond to search nodes, the merged representation of similar paths reduces the number of node expansions and thereby the overall search effort of WC-A\*pex. It is noteworthy that WC-A\*pex uses a different technique to speed up the search from existing bounded suboptimal search algorithms, most of which rely on node expansion orders to guide the search to quickly find a bounded suboptimal solution.

We empirically evaluate WC-A\*pex with different suboptimality bounds against competing algorithms on benchmark road networks with 1 to 14 million states and 2 to 34 million edges. The competing algorithms include WC-A\* and our adaptation of it to a bounded suboptimal variant using the same method provided by Cabrera et al. (2020), called WC-A\*-$\varepsilon$. WC-A\*-$\varepsilon$ is similar to WC-A\* but terminates the search immediately after the incumbent solution is proven to be $(1 + \varepsilon)$-suboptimal. Our experimental results show that WC-A\*pex significantly outperforms WC-A\* and WC-A\*-$\varepsilon$ although WC-A\* and WC-A\*-$\varepsilon$ are also based on BOSP algorithms. This, in turn, demonstrates the power of the merged representation of similar paths used in WC-A\*pex. Even with $\varepsilon = 0.01$ (i.e., with 1% suboptimality), WC-A\*pex achieves an order-of-magnitude speed-up over WC-A\* and WC-A\*-$\varepsilon$ on the largest road network. In comparison, WC-A\*-$\varepsilon$ for the same value of $\varepsilon$ achieves less than 20% runtime improvement over WC-A\*.

## 2 Terminology and Problem Definition

In this section, we formally define the WCSP and the BOSP problems. To set up a notation that serves the description of both problems, we define the cost of an edge as a pair of numbers. In the WCSP context, the first number indicates the cost, and the second number indicates the weight. In the BOSP context, both numbers represent the cost.

We use the **boldface** font to denote pairs and $p_i$, $i \in \{1, 2\}$, to denote the $i$-th component of a pair $\mathbf{p}$. The addition of two pairs $\mathbf{p}$ and $\mathbf{p}'$ is defined as $\mathbf{p} + \mathbf{p}' = (p_1 + p_1', p_2 + p_2')$. We say that $\mathbf{p}$ *(weakly) dominates* $\mathbf{p}'$, denoted as $\mathbf{p} \preceq \mathbf{p}'$, if $p_1 \leq p_1'$ and $p_2 \leq p_2'$. For an *approximation factor* (or, more precisely, a pair of approximation factors) $\boldsymbol{\varepsilon} = (\varepsilon_1, \varepsilon_2)$, we say that $\mathbf{p}$ $\boldsymbol{\varepsilon}$-*dominates* $\mathbf{p}'$, denoted as $\mathbf{p} \preceq_{\boldsymbol{\varepsilon}} \mathbf{p}'$, if $p_1 \leq (1 + \varepsilon_1) \cdot p_1'$ and $p_2 \leq (1 + \varepsilon_2) \cdot p_2'$.

A *(bi-objective) graph* is a tuple $G = \langle S, E, \mathbf{c} \rangle$, where $S$ is a finite set of *states* and $E \subseteq S \times S$ is a finite set of (directed) *edges*. $succ(s) = \{s' \in S : \langle s, s' \rangle \in E\}$ denotes the successors of state $s$. The *cost function* $\mathbf{c} : E \to \mathbb{R}_{\geq 0} \times \mathbb{R}_{\geq 0}$ maps an edge to its *cost*, which is a pair of non-negative

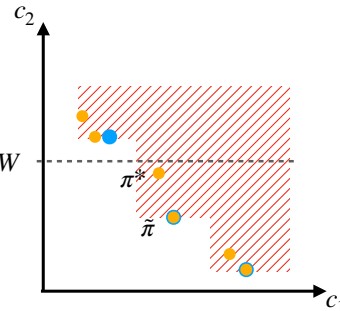

Figure 1: Example of the Pareto front (whose costs are shown by the orange dots) and an $(\varepsilon, 0)$-approximate Pareto front (whose costs are shown by the blue dots) from the start state to the goal state of a WCSP instance, respectively. The shaded region shows the costs that are $(\varepsilon, 0)$-dominated by at least one blue dot. Note that all orange dots are within the shaded region. Solutions $\pi^*$ and $\tilde{\pi}$ are an optimal solution and a $(1 + \varepsilon)$-suboptimal solution of the WCSP instance, respectively, with $\mathbf{c}(\tilde{\pi}) \preceq_{(\varepsilon, 0)} \mathbf{c}(\pi^*)$.

numbers. A *path* from state $s_1$ to state $s_\ell$ is a sequence of states $\pi = [s_1, s_2 \dots s_\ell]$ with $\langle s_j, s_{j+1} \rangle \in E$ for all $j = 1, 2 \dots \ell - 1$. $s_1 = s_{\text{start}}$ unless mentioned otherwise. Slightly abusing the notation, we define the cost of $\pi$ as $\mathbf{c}(\pi) = \sum_{j=1}^{\ell-1} \mathbf{c}(\langle s_j, s_{j+1} \rangle)$.

A *WCSP (problem) instance* is a tuple $P = \langle G, s_{\text{start}}, s_{\text{goal}}, W \rangle$, where $G$ is a graph, $s_{\text{start}} \in S$ is the *start state*, $s_{\text{goal}} \in S$ is the *goal state*, and $W \in \mathbb{R}_{>0}$ is the *weight limit*.[1] The two components of the cost function $\mathbf{c}$, $c_1$ and $c_2$, correspond to the cost and weight in the context of the WCSP problem, respectively. A path $\pi$ is a *solution* of $P$ if it is from $s_{\text{start}}$ to $s_{\text{goal}}$ and satisfies $c_2(\pi) \leq W$. We say that $P$ is *solvable* if it has a solution. An *optimal solution* of $P$ is a solution with the minimum $c_1$-value, denoted as $c_1^*$, of all solutions. Given a non-negative $\varepsilon$, a solution $\pi$ is $(1 + \varepsilon)$-*suboptimal* if $c_1(\pi) \leq (1 + \varepsilon) \cdot c_1^*$. A *bounded-suboptimal WCSP algorithm* takes a WCSP instance $P$ and a parameter $\varepsilon \geq 0$ as input and computes a $(1 + \varepsilon)$-suboptimal solution.

A path $\pi$ from $s_{\text{start}}$ to $s_{\text{goal}}$ is *Pareto-optimal* if there does not exist another path $\pi'$ from $s_{\text{start}}$ to $s_{\text{goal}}$ with $\mathbf{c}(\pi') \preceq \mathbf{c}(\pi)$ and $\mathbf{c}(\pi') \neq \mathbf{c}(\pi)$. The *Pareto front* $\Pi^*$ (from $s_{\text{start}}$ to $s_{\text{goal}}$) is the set of all Pareto-optimal paths. For a non-negative pair $\boldsymbol{\varepsilon}$, a set of paths $\Pi_{\boldsymbol{\varepsilon}}$ from $s_{\text{start}}$ to $s_{\text{goal}}$ is an $\boldsymbol{\varepsilon}$-*approximate Pareto front* from $s_{\text{start}}$ to $s_{\text{goal}}$ if, any path from $s_{\text{start}}$ to $s_{\text{goal}}$ is $\boldsymbol{\varepsilon}$-dominated by at least one path in $\Pi_{\boldsymbol{\varepsilon}}$. Note that different $\boldsymbol{\varepsilon}$-approximate Pareto fronts can exist for the same $s_{\text{start}}$, $s_{\text{goal}}$, and $\boldsymbol{\varepsilon}$.

A BOSP (problem) instance is a tuple $\langle G, s_{\text{start}}, s_{\text{goal}} \rangle$, where $G$ is a graph, $s_{\text{start}} \in S$ is the start state, and $s_{\text{goal}} \in S$ is the goal state. An *approximate BOSP algorithm* takes a BOSP instance and an approximation factor $\boldsymbol{\varepsilon}$ as input and computes an $\boldsymbol{\varepsilon}$-approximate Pareto front from $s_{\text{start}}$ to $s_{\text{goal}}$.

The following observation shows the connection between

---

[1]This $W$ is not to be confused with the $W$ used for the suboptimality bound in certain bounded suboptimal search algorithms.

a bounded-suboptimal WCSP algorithm and an approximate BOSP algorithm.

**Observation 1.** *For a solvable WCSP instance* $P = \langle G, s_{start}, s_{goal}, W \rangle$ *and* $\varepsilon \geq 0$, *any* $(\varepsilon, 0)$-*approximate Pareto front (from* $s_{start}$ *to* $s_{goal}$) $\Pi_\varepsilon$ *contains a* $(1 + \varepsilon)$-*suboptimal solution of* $P$.

*Proof.* Let $\pi^*$ denote an optimal solution of $P$. By definition, there exists a path $\pi \in \Pi_\varepsilon$ with $\mathbf{c}(\pi) \preceq_{(\varepsilon,0)} \mathbf{c}(\pi^*)$ (i.e., $c_1(\pi) \leq (1 + \varepsilon) \cdot c_1(\pi^*)$ and $c_2(\pi) \leq c_2(\pi^*) \leq W$). We can see that $\pi$ is a $(1 + \varepsilon)$-suboptimal solution of $P$. $\square$

See Figure 1 for a visualization of an $\varepsilon$-approximate Pareto front and a $(1+\varepsilon)$-suboptimal solution of a WCSP instance.

In this paper, we focus on heuristic-search-based WCSP algorithms. We assume that a *heuristic function* $\mathbf{h} : S \to \mathbb{R}_{\geq 0} \times \mathbb{R}_{\geq 0}$, which provides a lower bound on the cost from any given state $s$ to the goal state, is always available. Additionally, we assume that the heuristic function $\mathbf{h}$ is *consistent*, that is, $\mathbf{h}(s_{goal}) = \mathbf{0}$ and $\mathbf{h}(s) \preceq \mathbf{c}(e) + \mathbf{h}(s')$ for all $e = \langle s, s' \rangle \in E$. It is a common practice in existing WCSP and BOSP literature (Ahmadi et al. 2021, 2022b; Hernández et al. 2023; Zhang et al. 2022b) to use Dijkstra's algorithm (starting from $s_{goal}$) to compute the minimum cost $c_i^*(s)$ from any state $s$ to $s_{goal}$ for the $i$-th objective (while ignoring the other objective), $i = 1, 2$, and $\mathbf{h}(s) := (c_1^*(s), c_2^*(s))$ as the heuristic function. We call these heuristic functions *perfect-distance heuristics*.

## 3 Algorithmic Background

In this section, we review existing WCSP and BOSP algorithms, with a focus on BOA* (Hernández et al. 2023), WC-A* (Ahmadi et al. 2022b), and A*pex (Zhang et al. 2022b).

Many WCSP and BOSP algorithms such as BOA*, WC-A*, and A*pex follow the same *best-first bi-objective search* framework. In a best-first bi-objective search algorithm, a (search) node $n$ contains a state $s(n)$ and a $\mathbf{g}$-value $\mathbf{g}(n)$. The $\mathbf{f}$-value of $n$ is defined as $\mathbf{f}(n) = \mathbf{g}(n) + \mathbf{h}(s(n))$. The search algorithm maintains a priority queue $Open$, containing generated but not expanded nodes, and a set of *solutions*. $Open$ is initialized with a node that contains the start state $s_{start}$ and the $\mathbf{g}$-value $\mathbf{0}$. In each iteration, the algorithm extracts a node $n$ from $Open$ with an undominated $\mathbf{f}$-value of all nodes in $Open$. It performs a *dominance check* to determine whether $n$ or any of its descendants have the potential to be in the solution set. If not, it discards $n$. Otherwise, when $s(n) = s_{goal}$, the algorithm adds $n$ to the solution set or, when $s(n) \neq s_{goal}$, expands $n$ by generating a new node for each of the successor in $succ(s(n))$. The algorithm also performs a dominance check for each generated node and adds the generated node to $Open$ if it passes the dominance check. When $Open$ becomes empty, the algorithm terminates and returns the solution set.

Different best-first bi-objective search algorithms mainly differ in which information is contained in the nodes, which node is extracted from $Open$, and how the dominance checks work. Specifically, BOA*, WC-A*, and A*pex extract the node with the *lexicographically smallest* $\mathbf{f}$-value (i.e., extract the node with the smallest $f_1$-value and break

ties in favor of a smaller $f_2$-value) in each iteration. The dominance checks in both BOA* and A*pex check if the $\mathbf{f}$-value of a node is weakly dominated by the $\mathbf{f}$-value of any expanded node with the same state or $s_{goal}$.

### 3.1 BOA* and WC-A*

BOA* (Hernández et al. 2023) computes a Pareto front for the given start and goal states. In BOA*, each node $n$ corresponds to a path from $s_{start}$ to $s(n)$ with cost $\mathbf{g}(n)$. Hernández et al. (2023) show that, due to the consistent heuristic function BOA* uses, the $f_1$-values of extracted nodes are monotonically non-decreasing. Thus, BOA* only stores the minimum $g_2$-value of all expanded nodes for each state $s$ using a variable named $g_2^{\min}(s)$. Consequently, dominance checks can be done by checking if $g_2(n) < g_2^{\min}(s)$ and $f_2(n) < g_2^{\min}(s_{goal})$. This dominance check can be performed in constant time (in contrast to previous methods which required time linear in the number of nodes that reached $s$).

WC-A* is built on BOA* and computes an optimal solution for an input WCSP instance. It only maintains at most one *incumbent solution* in the solution set. In addition to the dominance checks of BOA*, WC-A* also discards nodes whose (1) $f_2$-values are larger than the weight limit $W$ or (2) $f_1$-values are not smaller than the $c_1$-value of the incumbent solution. Since WC-A* extracts nodes with monotonically non-decreasing $f_1$-values, it terminates (and returns the incumbent solution) once the minimum $f_1$-value in $Open$ is not smaller than the one of the incumbent solution. During the heuristic computation with Dijkstra's algorithm, the minimum-$c_1$ and minimum-$c_2$ paths from any state $s$ to $s_{goal}$ can also be obtained. We call these paths the *complementary paths* of $s$. When generating a node $n$, WC-A* tries to update the incumbent solution with better ones by connecting the corresponding path of $n$ with the complementary paths of $s(n)$.

WC-A* can be converted to a bounded suboptimal WCSP algorithm by terminating the algorithm when the minimum $f_1$-value in $Open$ is not smaller than the $f_1$-value of the incumbent solution divided by $(1 + \varepsilon)$. We denote this variant of WC-A* as WC-A*-$\varepsilon$ and include it in our empirical study.

It is noteworthy that Ahmadi et al. (2022b) propose WC-BA*, a bi-directional variant of WC-A* that runs two WC-A* searches (one starting from $s_{start}$ and the other starting from $s_{goal}$) concurrently. We omit WC-BA* in this paper because Ahmadi et al. (2022a) later show that WC-BA* does not dominate WC-A* in experimental results and, in fact, has larger average runtime in several scenarios.

### 3.2 A*pex

A*pex computes an $\varepsilon$-approximate Pareto front for the given start and goal states and a user-provided $\varepsilon$-value. In A*pex, a node is a so-called apex-path pair $\mathcal{AP} = \langle \mathbf{A}, \pi \rangle$ that consists of a cost pair $\mathbf{A}$, called the *apex*, and a path $\pi$, called the *representative path*. We define the $\mathbf{g}$-value of $\mathcal{AP}$ as $\mathbf{g}(\mathcal{AP}) := \mathbf{A}$ and $s(\mathcal{AP})$ as the last state of the representative path $\pi$. We have $\mathbf{f}(\mathcal{AP}) := \mathbf{g}(\mathcal{AP}) + \mathbf{h}(s(\mathcal{AP}))$ as the $\mathbf{f}$-value of $\mathcal{AP}$. Conceptually, an apex-path pair corresponds to a set of paths that end at the same state, and its apex is the

component-wise minimum value of the cost of these paths. $\mathcal{AP}$ is said to be $\varepsilon$-bounded if $\mathbf{c}(\pi) + \mathbf{h}(s(\mathcal{AP})) \preceq_{\boldsymbol{\varepsilon}} f(\mathcal{AP})$.

Whenever A*pex inserts an apex-path pair to $Open$ or the solution set, A*pex tries to *merge* apex-path pairs in $Open$ with the same state on condition that the resulting apex-path pair is $\varepsilon$-bounded. The implementation of A*pex by Zhang et al. (2022b) uses a list to maintain the apex-path pairs in $Open$ for each state and hence can iterate these apex-path pairs efficiently. When merging two apex-path pairs, the new apex is the component-wise minimum of the apexes of the two apex-path pairs, and the new representative path is either one of the two representative paths of the two apex-path pairs. See Figure 2(a) for a visualization of the two possible outcomes.

## 4  WC-A*pex

In this section, we describe WC-A*pex, our bounded sub-optimal WCSP algorithm that finds a $(1+\varepsilon)$-suboptimal solution for a user-provided $\varepsilon$. We first describe the base algorithm of WC-A*pex and then provide its theoretical results and speed-up techniques.

Observation 1 shows that a $(1+\varepsilon)$-suboptimal solution of a WCSP instance can be found in a corresponding $(\varepsilon, 0)$-approximate Pareto front. This motivates us to propose WC-A*pex, which can be viewed as a modified A*pex with $\boldsymbol{\varepsilon} = (\varepsilon, 0)$ and additional prunings. Note that we use bold-face $\boldsymbol{\varepsilon}$ and regular $\varepsilon$ to distinguish between the approximation factor that A*pex would use and the user-provided parameter for the WCSP problem. Similarly to A*pex, a node in WC-A*pex is an apex-path pair $\mathcal{AP} = \langle \mathbf{A}, \pi \rangle$. Since the second component of $\boldsymbol{\varepsilon}$ is set to 0, when merging two apex-path pairs $\langle \mathbf{A}, \pi \rangle$ and $\langle \mathbf{A}', \pi' \rangle$ with $c_2(\pi) < c_2(\pi')$, WC-A*pex cannot choose the $\pi'$ as the new representative path. Otherwise, the resulting apex path pair is not $\boldsymbol{\varepsilon}$-bounded. See Figure 2(b) for a visualization of merging two apex-path pairs in WC-A*pex. In addition to dominance checks, WC-A*pex also prunes nodes whose $f_2$-values are larger than $W$.

Algorithm 1 shows the pseudocode of WC-A*pex. It starts with a single apex-path pair $\langle \mathbf{0}, [s_{\text{start}}] \rangle$ in $Open$ (Line 1). At each iteration, WC-A*pex extracts an apex-path pair from $Open$ with the lexicographically smallest $\mathbf{f}$-value (Line 5). Same as BOA*, WC-A*pex maintains a $g_2^{\min}(s)$ for each state $s$ that contains the smallest $g_2$-value of all expanded nodes with state $s$ by updating it on Line 10. Both after extracting (that is, after Line 5) and before generating (that is, before Line 16) an apex-path pair $\mathcal{AP}$ with state $s$, WC-A*pex discards the apex-path pair if (1) $g_2(\mathcal{AP}) \geq g_2^{\min}(s(\mathcal{AP}))$ or (2) $f_2(\mathcal{AP}) > W$. Case (1) holds iff there exists an expanded node with state $s$ whose $\mathbf{g}$-value weakly dominated $\mathbf{g}(\mathcal{AP})$, which implies that any solution found via $\mathcal{AP}$ is also $(\varepsilon, 0)$-dominated by a solution found via expanded node and hence $\mathcal{AP}$ can be safely discarded. In Case (2), $\mathcal{AP}$ is pruned since the representative path of $\mathcal{AP}$ cannot be extended to a solution (whose $c_2$-value needs to be not larger than $W$).

When WC-A*pex expands an apex-path pair $\mathcal{AP}$ with state $s$, it generates a child apex-path pair for each successor $s'$ of state $s$. The apex of the child apex-path pair is the

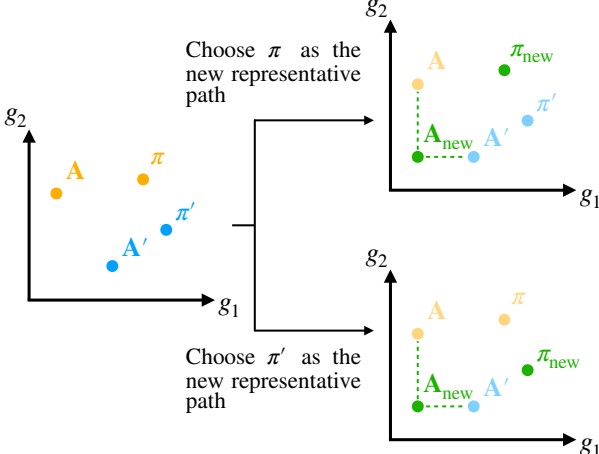

Choose $\pi$ as the new representative path

Choose $\pi'$ as the new representative path

(a) A*pex (adapted from Figure 2 by Zhang et al. (2022b))

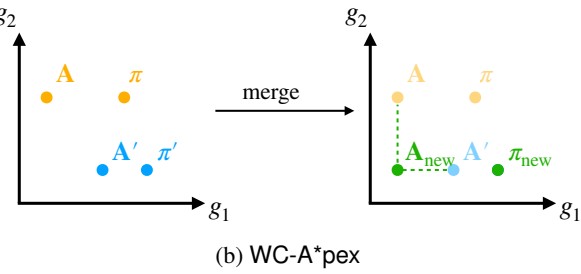

merge

(b) WC-A*pex

Figure 2: Examples of merging apex-path pairs $\langle \mathbf{A}, \pi \rangle$ (orange) and $\langle \mathbf{A}', \pi' \rangle$ (blue) into apex-path pair $\langle \mathbf{A}_{\text{new}}, \pi_{\text{new}} \rangle$ (green) in A*pex and WC-A*pex, respectively.

sum of the apex of $\mathcal{AP}$ and $\mathbf{c}(\langle s, s' \rangle)$ (Line 12), and the representative path of the child apex-path pair is the representative path of $\mathcal{AP}$ appended with state $s'$ (Line 13). Before adding the child apex-path pair $\mathcal{AP}'$ to $Open$, WC-A*pex attempts to merge the apex-path pair with an apex-path pair in $Open[s(\mathcal{AP}')]$ on condition that the resulting apex-path pair is $(\varepsilon, 0)$-bounded (Lines 19-25), where $\varepsilon$ is the input suboptimality factor and $Open[s(\mathcal{AP}')]$ denotes the set of apex-path pairs in $Open$ with state $s(\mathcal{AP}')$.

WC-A*pex terminates when it finds a solution (Line 9) or $Open$ becomes empty (Line 17), in which case, there is no solution for the given WCSP instance.

We use an example WCSP instance to demonstrate how WC-A*pex works. Figure 3(a) shows the example graph. The weight limit $W$ and $\varepsilon$ are set to 7 and 0.2, respectively. Figure 3(b) shows the costs of all paths from $s_{\text{start}}$ to $s_{\text{goal}}$ in this graph. We can see that path $[s_{\text{start}}, s_1, s_3, s_4, s_{\text{goal}}]$, whose cost is $(7, 7)$, is the optimal solution for this example. Moreover, since the second best path $[s_{\text{start}}, s_1, s_3, s_4, s_{\text{goal}}]$ has a large $c_1$-value of 13, $[s_{\text{start}}, s_1, s_3, s_4, s_{\text{goal}}]$ is also the only 1.2-suboptimal solution for this example. We use the perfect-distance heuristic. In the text below, slightly abusing the notation, we use subscript $i$ to index an apex-path pair and tuple $\langle s(\mathcal{AP}_i), \mathbf{f}(\mathcal{AP}_i), \mathbf{c}(\pi_i) \rangle$ to denote an apex-path

## Algorithm 1: WC-A*pex

**Input** : $P = \langle G, s_{\text{start}}, s_{\text{goal}}, W \rangle$
        $\varepsilon$
        **h**

1   $Open \leftarrow \{\langle \mathbf{0}, [s_{\text{start}}] \rangle\}$
2   **for each** $s \in S$ **do**
3     $g_2^{\min}(s) \leftarrow \infty$
4   **while** $Open \neq \emptyset$ **do**
5     $\mathcal{AP} = \langle \mathbf{A}, \pi \rangle \leftarrow Open.\text{extract\_min}()$
6     **if** $g_2(\mathcal{AP}) \geq g_2^{\min}(s(\mathcal{AP})) \vee f_2(\mathcal{AP}) > W$ **then**
7       **continue**
8     **if** $s(\mathcal{AP}) = s_{goal}$ **then**
9       **return** $\pi$
10    $g_2^{\min}(s(\mathcal{AP})) \leftarrow g_2(\mathcal{AP})$
11    **for** $s' \in succ(s(\mathcal{AP}))$ **do**
12      $\mathbf{A}' \leftarrow \mathbf{A} + \mathbf{c}(\langle s(\mathcal{AP}), s' \rangle)$
13      $\pi' \leftarrow \pi.\text{append}(s')$
14      **if** $A'_2 \geq g_2^{\min}(s') \vee A'_2 + h_2(s') > W$ **then**
15        **continue**
16      insert\_to\_Open($\langle \mathbf{A}', \pi' \rangle$)
17 **return** *None*

18 **Function** *insert\_to\_Open*($\mathcal{AP}' = \langle \mathbf{A}', \pi' \rangle$)**:**
19    **for** $\mathcal{AP} = \langle \mathbf{A}, \pi \rangle \in Open[s(\mathcal{AP}')]$ **do**
20      $\mathbf{A}_{\text{new}} \leftarrow (\min(A_1, A'_1), \min(A_2, A'_2))$
21      $\pi_{\text{new}} \leftarrow$ the one of $\pi$ and $\pi'$ with the smaller $c_2$-value, breaking ties in favor of a smaller $c_1$-value
22      **if** $\langle \mathbf{A}_{new}, \pi_{new} \rangle$ is $(\varepsilon, 0)$-*bounded* **then**
23        remove $\mathcal{AP}$ from $Open$
24        add $\langle \mathbf{A}_{\text{new}}, \pi_{\text{new}} \rangle$ to $Open$
25        **return**
26    add $\mathcal{AP}'$ to $Open$
27    **return**

pair $\mathcal{AP}_i = \langle \mathbf{A}_i, \pi_i \rangle$.

- In Iteration 1, WC-A*pex expands apex-path pair $\mathcal{AP}_1 = \langle s_{\text{start}}, (5, 5), (0, 0) \rangle$ and generates two child apex-path pairs $\mathcal{AP}_2 = \langle s_1, (5, 5), (1, 2) \rangle$ and $\mathcal{AP}_3 = \langle s_2, (6, 6), (3, 2) \rangle$.

- In Iteration 2, WC-A*pex expands apex-path pair $\mathcal{AP}_2$ and generates two child apex-path pairs $\mathcal{AP}_4 = \langle s_2, (5, 7), (2, 3) \rangle$ and $\mathcal{AP}_5 = \langle s_3, (7, 5), (5, 3) \rangle$. $\mathcal{AP}_4$ is merged with $\mathcal{AP}_3$ in $Open$, resulting in apex-path pair $\mathcal{AP}_6 = \langle s_2, (5, 6), (3, 2) \rangle$. $\mathcal{AP}_6$ is $(\varepsilon, 0)$-bounded because $(3, 2) + \mathbf{h}(s_2) = (6, 6) \preceq_{(\varepsilon, 0)} (5, 6)$.

- In Iteration 3, WC-A*pex expands apex-path pair $\mathcal{AP}_6$ and generates child apex-path pair $\mathcal{AP}_7 = \langle s_3, (5, 6), (4, 4) \rangle$. $\mathcal{AP}_7$ is not merged with $\mathcal{AP}_5$ because, given that the new representative path would have a cost of $(5, 3)$ and the new $\mathbf{f}$-value would be $(5, 5)$, $(5, 3) + \mathbf{h}(s_3) = (7, 5)$ does not $(\varepsilon, 0)$-dominate $(5, 5)$.

- In Iteration 4, WC-A*pex expands apex-path pair $\mathcal{AP}_7$ and generates two child apex-path pairs $\mathcal{AP}_8 = \langle s_4, (5, 8), (5, 6) \rangle$ and $\mathcal{AP}_9 = \langle s_5, (13, 6), (9, 5) \rangle$. $\mathcal{AP}_8$ is pruned because $f_2(\mathcal{AP}_8) > W$.

- In Iteration 5, WC-A*pex expands apex-path pair $\mathcal{AP}_5$ and generates two child apex-path pairs $\mathcal{AP}_{10} =$

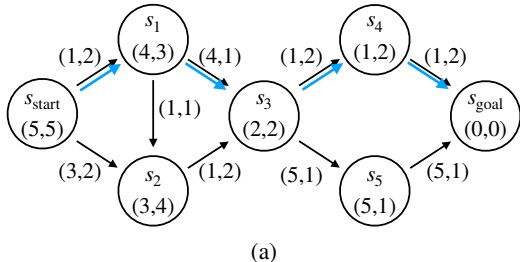

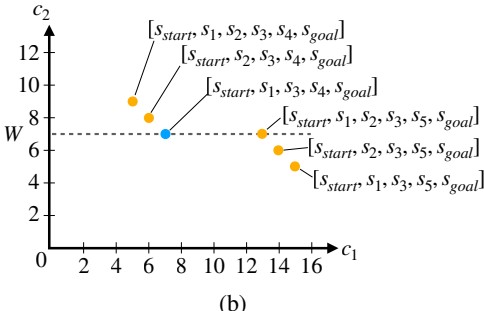

Figure 3: An example WCSP instance. (a) shows the graph for this WCSP instance, where the pair of numbers inside each state is its **h**-value and the blue arrows show the optimal solution for weight limit $W = 7$. (b) shows the costs of all paths from $s_{\text{start}}$ to $s_{\text{goal}}$ in the graph.

$\langle s_4, (7, 7), (6, 5) \rangle$ and $\mathcal{AP}_{11} = \langle s_5, (15, 5), (10, 4) \rangle$. $\mathcal{AP}_{11}$ is merged with $\mathcal{AP}_9$ in $Open$, resulting in apex-path pair $\mathcal{AP}_{12} = \langle s_5, (13, 5), (10, 4) \rangle$. $\mathcal{AP}_{12}$ is $(\varepsilon, 0)$-bounded because $(10, 4) + \mathbf{h}(s_5) = (15, 5) \preceq_{(\varepsilon, 0)} (13, 5)$.

- In Iteration 6, WC-A*pex expands apex-path pair $\mathcal{AP}_{10}$ and generates child apex-path pair $\mathcal{AP}_{13} = \langle s_{\text{goal}}, (7, 7), (7, 7) \rangle$.

- In Iteration 7, WC-A*pex expands apex-path pair $\mathcal{AP}_{13}$ and returns a solution with cost $(7, 7)$.

In this example, WC-A*pex finds the optimal solution. Two merges happen during the entire process, which are in Iteration 2 between $\mathcal{AP}_3 = \langle s_2, (6, 6), (3, 2) \rangle$ and $\mathcal{AP}_4 = \langle s_2, (5, 7), (2, 3) \rangle$ and in Iteration 5 between $\mathcal{AP}_9 = \langle s_5, (13, 6), (9, 5) \rangle$ and $\mathcal{AP}_{11} = \langle s_5, (15, 5), (10, 4) \rangle$, respectively. The representative paths of $\mathcal{AP}_3$ and $\mathcal{AP}_4$ are $\pi_3 = [s_{\text{start}}, s_2]$ and $\pi_4 = [s_{\text{start}}, s_1, s_2]$, respectively. Compared to $\mathcal{AP}_5$, which is expanded in Iteration 5 and eventually extended to the solution, $\mathcal{AP}_3$ and $\mathcal{AP}_4$ have lexicographically smaller $\mathbf{f}$-values and appear to be more promising. However, the two possible extensions of $\mathcal{AP}_3$ and $\mathcal{AP}_4$ to $s_{\text{goal}}$ either violate the weight limit (via $[s_2, s_3, s_4, s_{\text{goal}}]$) or have large $c_1$-values (via $[s_2, s_3, s_5, s_{\text{goal}}]$), as the algorithm finds out in Iterations 3-4. Without merging, other existing WCSP algorithms, like WC-A*, would represent $\pi_3$ and $\pi_4$ as two different nodes and spend more search effort.

## 4.1 Theoretical Results

In this section, we show that WC-A*pex always returns a $(1 + \varepsilon)$-suboptimal solution, given a solvable WCSP instance. Note that we say a WCSP instance is solvable if there exists a solution for this WCSP instance (Sec. 2).

**Lemma 1.** *If we have $g_2(\mathcal{AP}) \geq g_2^{min}(s(\mathcal{AP}))$ on Line 6 or 14, there exists an expanded apex-path pair $\mathcal{AP}'$ with state $s(\mathcal{AP}') = s(\mathcal{AP})$ and $\mathbf{f}(\mathcal{AP}') \preceq_{(\varepsilon,0)} \mathbf{f}(\mathcal{AP})$.*

*Proof.* This lemma is rephrased from Lemma 2 by Zhang et al. (2022b), and the same proof applies. ☐

For the rest of this section, we use $\pi^* = [s_1^*, s_2^* \ldots s_\ell^*]$ to denote an optimal solution for the given WCSP instance, assuming that a solution exists. Note that we have $s_\ell^* = s_{\text{goal}}$. We use $\pi_j^* = [s_1^*, s_2^* \ldots s_j^*], j = 1, 2 \ldots \ell$, to denote the subpath of $\pi^*$ that contains the first $j$ states.

**Lemma 2.** *We assume that the given WCSP instance is solvable. At the beginning of each iteration, for any expanded apex-path pair $\mathcal{AP}$, if there exists $j \in \{1, 2 \ldots \ell - 1\}$ that satisfies $s(\mathcal{AP}) = s_j^*$ and $\mathbf{g}(\mathcal{AP}) \preceq \mathbf{c}(\pi_j^*)$, there exists an apex-path $\mathcal{AP}' \in Open$ and $k > j$ that satisfy $s(\mathcal{AP}') = s_k^*$ and $\mathbf{g}(\mathcal{AP}') \preceq \mathbf{c}(\pi_k^*)$.*

*Proof.* We prove this lemma by induction on $j$, starting from $j = \ell - 1$ and going backward. Consider an expanded apex-pair $\mathcal{AP}$ with $s(\mathcal{AP}) = s_{\ell-1}^*$ and $\mathbf{g}(\mathcal{AP}) \preceq \mathbf{c}(\pi_{\ell-1}^*)$. When it is expanded, WC-A*pex generates a child apex-path pair $\mathcal{AP}'$ with state $s_\ell^* = s_{\text{goal}}$. We have $g_2^{\min}(s_{\text{goal}}) = \infty$ because, if $s(\mathcal{AP}) = s_{\text{goal}}$, the algorithm terminates on Line 9 and cannot reach Line 10 to update $g_2^{\min}(s_{\text{goal}})$. We have $\mathbf{g}(\mathcal{AP}') = \mathbf{g}(\mathcal{AP}) + \mathbf{c}(\langle s_{\ell-1}^*, s_\ell^* \rangle) \preceq \mathbf{c}(\pi^*)$. Since the heuristic $\mathbf{h}$ is consistent, we have $\mathbf{h}(s_\ell^*) = \mathbf{0}$ and hence $f_2(\mathcal{AP}') \leq c_2(\pi^*) \leq W$. Therefore, $\mathcal{AP}'$ is not pruned on Line 15 and then inserted into $Open$. If $\mathcal{AP}'$ has been extracted from $Open$, the algorithm should have terminated. We show that $\mathcal{AP}'$ is still in $Open$ for the current iteration.

Now we assume the lemma holds for $j = i + 1$. Consider an expanded apex-path pair $\mathcal{AP}$ with $s(\mathcal{AP}) = s_i^*$ and $\mathbf{g}(\mathcal{AP}) \preceq \mathbf{c}(\pi_i^*)$. When expanding $\mathcal{AP}$, one of the child apex-path pairs, denoted as $\mathcal{AP}'$, is generated with state $s_{i+1}^*$. We have $\mathbf{g}(\mathcal{AP}') = \mathbf{g}(\mathcal{AP}) + \mathbf{c}(\langle s_i^*, s_{i+1}^* \rangle) \preceq \mathbf{c}(\pi_i^*) + \mathbf{c}(\langle s_i^*, s_{i+1}^* \rangle) = \mathbf{c}(\pi_{i+1}^*)$. We distinguish two cases:

1. $\mathcal{AP}'$ is pruned on Line 15. Since $f_2(\mathcal{AP}') = g_2(\mathcal{AP}') + h_2(s_{i+1}^*) \leq c_2(\pi^*) \leq W$, we must have $g_2^{\min}(s_{i+1}^*) \leq g_2(\mathcal{AP}')$. From Lemma 1, there exists an expanded apex-path pair with state $s_{i+1}^*$ and whose $\mathbf{g}$-value weakly dominates $\mathbf{g}(\mathcal{AP}')$ and hence $\mathbf{c}(\pi_{i+1}^*)$. Because the lemma holds for $j = i+1$, there exists an apex-path pair $\mathcal{AP}'' \in Open$ with $s(\mathcal{AP}'') = s_k^*$ and $\mathbf{g}(\mathcal{AP}'') \preceq \mathbf{c}(\pi_k^*)$ for some $k > i + 1$. The lemma holds for $j = i$.

2. $\mathcal{AP}'$ is not pruned on Line 15 and is then inserted to $Open$ with or without merging with another apex-path pair. In either case, an apex-path pair with state $s_{i+1}^*$ and whose $\mathbf{g}$-value weakly dominates $\mathbf{c}(\pi_{i+1}^*)$ is inserted to $Open$. The lemma holds if this new apex-path pair is

still in $Open$. If this new apex-path pair has been extracted, either it is pruned or expanded, there is an expanded apex-path pair with state $s_{i+1}^*$ and whose $\mathbf{g}$-value weakly dominates $\mathbf{c}(\pi_{i+1}^*)$. Because the lemma holds for $j = i + 1$, the lemma also holds for $j = i$.

Therefore, the lemma holds for all $j \in \{1, 2 \ldots \ell - 1\}$. ☐

**Theorem 1.** *WC-A*pex returns a $(1 + \varepsilon)$-suboptimal solution, given a solvable WCSP problem instance.*

*Proof.* In the first iteration, there is one apex-path pair, denoted as $\mathcal{AP}_{\text{init}}$, in $Open$. Note that we have $s(\mathcal{AP}_{\text{init}}) = s_0^*$ and $\mathbf{g}(\mathcal{AP}_{\text{init}}) \preceq \mathbf{c}(\pi_0^*)$. $\mathcal{AP}_{\text{init}}$ is then expanded in the first iteration, and, from Lemma 2, $Open$ always contains at least one apex-path pair at the beginning of all future iterations. Therefore, WC-A*pex will not reach Line 17 and returns $None$. Let $\pi_{\text{sol}}$ and $\mathcal{AP}_{\text{sol}}$ denote the path returned by WC-A*pex and the apex-path pair that contains $\pi_{\text{sol}}$, respectively. $\pi_{\text{sol}}$ is a solution because it ends at state $s_{\text{goal}}$ and its $c_2$-value is not larger than $W$ (otherwise $\mathcal{AP}_{\text{sol}}$ is pruned on Line 7). Because $\mathcal{AP}_{\text{sol}}$ is $(\varepsilon, 0)$-bounded, we have $(1 + \varepsilon) \cdot f_1(\mathcal{AP}_{\text{sol}}) \geq c_1(\pi_{\text{sol}})$. If $\pi_{\text{sol}}$ is not $(1 + \varepsilon)$-suboptimal, i.e., $c_1(\pi_{\text{sol}}) > (1 + \varepsilon) \cdot c_1(\pi^*)$, we have $f_1(\mathcal{AP}_{\text{sol}}) > c_1(\pi^*)$. From Lemma 2, there always exists an apex-path pair $\mathcal{AP} \in Open$ and $j$ with $s(\mathcal{AP}) = s_j^*$ and $\mathbf{g}(\mathcal{AP}) \preceq \mathbf{c}(\pi_j^*)$. We have $\mathbf{f}(\mathcal{AP}) = \mathbf{g}(\mathcal{AP}) + \mathbf{h}(s(\mathcal{AP})) \preceq \mathbf{c}(\pi_j^*) + \mathbf{h}(s_j^*) \preceq \mathbf{c}(\pi^*)$. The algorithm must extract $\mathcal{AP}$ before extracting $\mathcal{AP}_{\text{sol}}$, and hence we find a contradiction. Therefore, $\pi_{\text{sol}}$ must be a $(1 + \varepsilon)$-suboptimal solution. ☐

## 4.2 Speed-up Techniques

In this section, we describe some speed-up techniques for an efficient implementation of WC-A*pex. Some of these techniques are also used by existing algorithms like WC-A*, and hence we omit the theoretical results for them.

**Efficient data structures:** Same as WC-A*, we use a bucket queue to implement $Open$ for WC-A*pex. Additionally, for each state $s$, we use a doubly linked list to keep track of all apex-path pairs in $Open$ with state $s$. Therefore, WC-A*pex can efficiently iterate over $Open[s]$ for any given state $s$ when checking for merging on Lines 19-25 and efficiently update the doubly linked list when an apex-path pair is extracted from or inserted into $Open$. Our preliminary results showed that these improved data structures speed up the original implementation of A*pex by an order of magnitude.

**Early solution update:** Similar to WC-A*, we can maintain and update an incumbent solution in WC-A*pex using complementary paths. An apex-path pair $\mathcal{AP}$ is pruned if $(1 + \varepsilon) \cdot f_1(\mathcal{AP})$ is not smaller than the $c_1$-value of the incumbent solution. With early solution update, WC-A*pex terminates when $Open$ becomes empty and returns the incumbent solution.

| Road network | $\varepsilon$ | WC-A* (-$\varepsilon$) Runtime Avg. | Max | #exp Avg. | Max | WC-A*pex Runtime Avg. | Max | #exp Avg. | Max | Speed-up |
|---|---|---|---|---|---|---|---|---|---|---|
| FLA | 0 | 0.17 | 4.48 | 2,397K | 49,400K | | | | | |
| | 0.01 | 0.16 | 5.12 | 2,135K | 47,994K | 0.05 | 0.79 | 274K | 3,543K | 2.95 |
| | 0.05 | 0.10 | 3.75 | 1,446K | 41,380K | 0.03 | 0.53 | 158K | 2,290K | 2.96 |
| | 0.10 | 0.07 | 4.48 | 905K | 41,380K | 0.02 | 0.50 | 95K | 2,128K | 3.12 |
| | 0.20 | 0.03 | 2.32 | 350K | 18,578K | 0.01 | 0.31 | 37K | 1,291K | 2.36 |
| NE | 0 | 0.36 | 8.38 | 3,631K | 59,122K | | | | | |
| | 0.01 | 0.30 | 5.28 | 3,051K | 44,032K | 0.09 | 2.56 | 473K | 8,302K | 3.20 |
| | 0.05 | 0.14 | 3.01 | 1,621K | 26,694K | 0.04 | 0.48 | 203K | 2,461K | 3.85 |
| | 0.10 | 0.05 | 0.86 | 694K | 9,610K | 0.02 | 0.18 | 96K | 1,160K | 2.48 |
| | 0.20 | 0.01 | 0.40 | 146K | 6,914K | 0.01 | 0.13 | 24K | 810K | 0.99 |
| LKS | 0 | 9.04 | 129.53 | 62,583K | 721,867K | | | | | |
| | 0.01 | 7.47 | 97.93 | 52,473K | 551,673K | 1.12 | 15.76 | 3,969K | 40,169K | 6.65 |
| | 0.05 | 4.14 | 60.09 | 29,640K | 327,693K | 0.34 | 3.44 | 1,504K | 11,486K | 12.30 |
| | 0.10 | 1.85 | 38.34 | 14,569K | 200,622K | 0.19 | 2.87 | 889K | 9,306K | 9.63 |
| | 0.20 | 0.15 | 4.09 | 1,938K | 31,143K | 0.04 | 0.44 | 171K | 2,062K | 3.62 |
| E | 0 | 10.88 | 138.31 | 71,758K | 763,176K | | | | | |
| | 0.01 | 9.16 | 132.30 | 61,861K | 715,655K | 1.58 | 16.76 | 5,246K | 51,904K | 5.81 |
| | 0.05 | 5.22 | 76.86 | 37,228K | 473,144K | 0.38 | 3.73 | 1,667K | 14,754K | 13.73 |
| | 0.10 | 1.94 | 40.36 | 16,331K | 279,604K | 0.21 | 3.17 | 955K | 12,604K | 9.09 |
| | 0.20 | 0.42 | 21.54 | 3,518K | 138,240K | 0.05 | 1.59 | 216K | 6,464K | 8.12 |
| W | 0 | 9.03 | 227.76 | 64,918K | 1,096,261K | | | | | |
| | 0.01 | 7.75 | 180.26 | 59,166K | 920,302K | 1.02 | 32.27 | 4,181K | 86,746K | 7.56 |
| | 0.05 | 4.57 | 70.20 | 38,849K | 553,971K | 0.51 | 7.10 | 2,286K | 21,667K | 8.89 |
| | 0.10 | 1.91 | 57.56 | 19,629K | 467,409K | 0.26 | 3.60 | 1,251K | 16,178K | 7.27 |
| | 0.20 | 0.57 | 41.62 | 6,242K | 304,128K | 0.11 | 2.81 | 448K | 12,997K | 5.32 |
| CAL | 0 | 0.76 | 21.48 | 7,717K | 138,645K | | | | | |
| | 0.01 | 0.66 | 20.42 | 6,890K | 131,590K | 0.21 | 4.50 | 998K | 16,659K | 3.13 |
| | 0.05 | 0.33 | 6.52 | 4,027K | 54,052K | 0.10 | 1.41 | 504K | 5,771K | 3.24 |
| | 0.10 | 0.19 | 3.99 | 2,427K | 46,152K | 0.06 | 1.17 | 314K | 5,771K | 2.86 |
| | 0.20 | 0.06 | 3.38 | 999K | 38,419K | 0.04 | 1.09 | 143K | 5,325K | 1.84 |
| CTR | 0 | 34.12 | 243.42 | 180,893K | 1,106,707K | | | | | |
| | 0.01 | 28.58 | 222.59 | 158,448K | 1,013,655K | 2.99 | 30.04 | 8,431K | 67,054K | 9.55 |
| | 0.05 | 17.15 | 185.34 | 96,893K | 804,131K | 1.10 | 11.75 | 3,597K | 32,594K | 15.62 |
| | 0.10 | 8.16 | 99.93 | 47,943K | 468,607K | 0.65 | 7.38 | 2,066K | 18,877K | 12.50 |
| | 0.20 | 1.16 | 39.79 | 8,979K | 233,685K | 0.18 | 3.24 | 476K | 10,624K | 6.52 |

Table 1: Average and maximum runtimes (in seconds), average and maximum node expansions, and speed-ups of WC-A*pex over WC-A*-$\varepsilon$ in average runtimes for instances on different road networks.

## 5 Experimental Results

In this section, we evaluate WC-A*pex with WCSP instances on road networks from the 9th DIMACS Implementation Challenge: Shortest Path.[2] We investigate WC-A*pex with different $\varepsilon$-values and compare the runtime and node expansions of WC-A*, WC-A*-$\varepsilon$, and WC-A*pex. We implemented WC-A*pex in C++[3]. We used the C++ implementation of WC-A* provided by the original authors[4] and implemented WC-A*-$\varepsilon$ based on it.

We choose nine road networks FLA (1.1M states and 2.7M edges), NE (1.5M states and 3.9M edges), CAL (1.9M states and 4.7M edges), LKS (2.8M states and 6.9M edges), E (3.6M states and 8.8M edges), W (6.3M states and 15.2M edges), and CTR (14.1M states and 34.3M edges) from the DIMACS data set. The $c_1$- and $c_2$-values for each edge are the travel time and distance, respectively, both available from the DIMACS data set. In other words, each WCSP instance corresponds to computing a path that is bounded-suboptimal in terms of travel time and with travel distance no larger than a specified limit. For each road network, we use the same 100 $s_{\text{start}}$ and $s_{\text{goal}}$ pairs used by Sedeño-Noda and Colebrook (2019) and Ahmadi et al. (2021). Following the literature (Cabrera et al. 2020; Ahmadi et al. 2022b), for each $s_{\text{start}}$ and $s_{\text{goal}}$ pair, we generate a WCSP instance with the weight limit $W = c_2^{\text{lb}} + \delta \cdot (c_2^{\text{ub}} - c_2^{\text{lb}})$ based on a *tightness factor* $\delta$, where $c_2^{\text{lb}}$ and $c_2^{\text{ub}}$ are the minimum and maximum $c_2$-values of Pareto-optimal paths from $s_{\text{start}}$ to $s_{\text{goal}}$, respectively. Conceptually, a smaller $\delta$-value corresponds to a tighter weight limit. For each $s_{\text{start}}$ and $s_{\text{goal}}$ pair, we use three tightness factors $0.25$, $0.5$, and $0.75$. Therefore, we have 300 WCSP instances for each road network.

For each WCSP instance, we evaluate WC-A*-$\varepsilon$ and WC-

[2]http://www.diag.uniroma1.it/challenge9/download.shtml.

[3]Upon acceptance, the code will be made publicly available.

[4]https://bitbucket.org/s-ahmadi/biobj/src/master/.

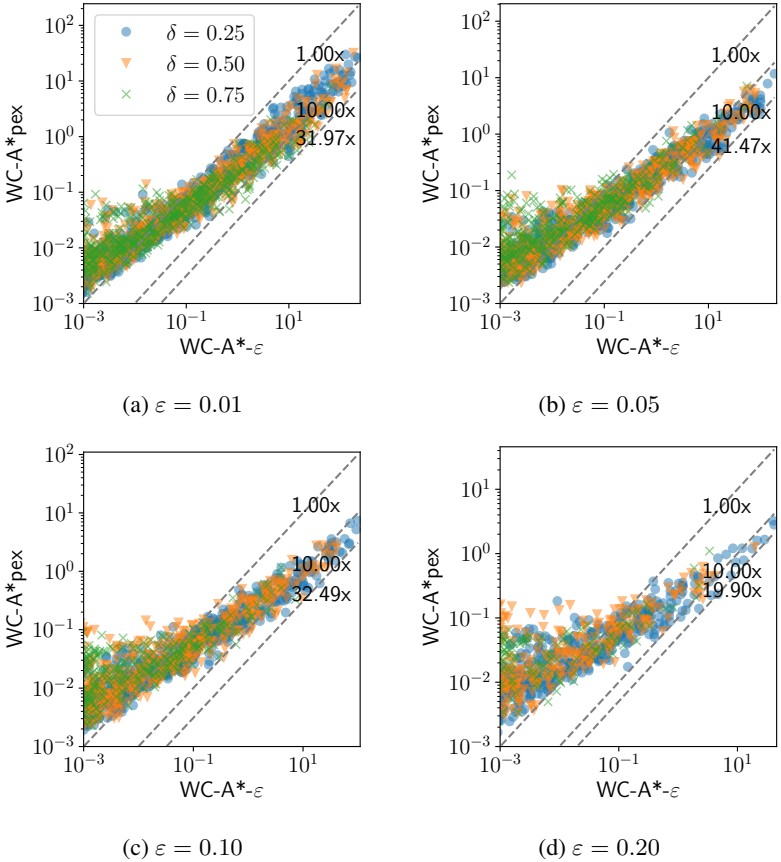

(a) $\varepsilon = 0.01$            (b) $\varepsilon = 0.05$

(c) $\varepsilon = 0.10$            (d) $\varepsilon = 0.20$

Figure 4: Runtimes of WC-A*-$\varepsilon$ and WC-A*pex on all WCSP instances with different suboptimality factors.

A*pex with three $\varepsilon$-values 0.01, 0.05, and 0.1. Table 1 shows the average and maximum runtimes (in seconds) and node expansions (#exp) of WC-A*, WC-A*-$\varepsilon$, and WC-A*pex over all WCSP instances. The results for WC-A* are shown in the rows with $\varepsilon = 0$. With $\varepsilon = 0.01$, i.e., a 1% suboptimality allowed, WC-A*pex solved the WCSP instances on the largest road network (CTR) faster than WC-A* by more than $11\times$ on average. However, the speed-up of WC-A*-$\varepsilon$ with $\varepsilon = 0.01$ compared to WC-A* is only less than 20% on average. This is because WC-A*-$\varepsilon$ still needs to expand a large number of nodes to prove an incumbent solution is bounded-suboptimal. The runtimes and node expansions of WC-A*pex are always smaller than the ones of WC-A* and WC-A*-$\varepsilon$ with the same $\varepsilon$-value on all road networks, which shows that merging apex-path pairs greatly reduce the runtimes and node expansions.

Figure 4 shows the individual runtimes (in seconds) of WC-A*pex and WC-A*-$\varepsilon$ for all WCSP instances and $\varepsilon$-values. We use different markers for different $\delta$-values that are used to generate the WCSP instances. The diagonal dashed lines and the numbers along them denote different speed-ups ($1\times$, $10\times$, and the maximum speed-up) of WC-A*pex over WC-A*-$\varepsilon$ . For different $\delta$-values, WC-A*pex shows a similar trend regarding the speed-ups over WC-

A*-$\varepsilon$. Although WC-A*pex were slower than WC-A*-$\varepsilon$ on easy instances (which both algorithm solved mostly within around 0.1 second), in more time-consuming instances (represented by the points on the top-right corners), WC-A*pex achieved significant speed-ups over WC-A*-$\varepsilon$.

## 6 Conclusions

In this paper, we proposed a bounded suboptimal WCSP algorithm called WC-A*pex. WC-A*pex is built on A*pex, a state-of-the-art approximate BOSP algorithm. It computes a $(1 + \varepsilon)$-suboptimal path, for a user-specified $\varepsilon$. Its empirical performance on benchmark road networks highlights two important computational aspects of it. First, huge gains in runtime efficiency—up to an order of magnitude—are possible with only a small compromise—within 1% suboptimality—on the cost of the solution. Second, the use of the merged representations of paths with similar costs and weights reduces the number of node expansions and is critical to the success of WC-A*pex over WC-A* and WC-A*-$\varepsilon$.

In future work, we intend to generalize WC-A*pex to the case of multiple costs, multiple weights (Skyler et al. 2022), or both. Another direction is to enhance WC-A*pex with recent algorithmic advancements (Zhang et al. 2022a) and develop it towards an anytime algorithm.

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
