# OpenReview forum: "Bounded Suboptimal Weight-Constrained Shortest-Path Search via Efficient Representation of Paths"
_icaps-conference.org/ICAPS/2024/Conference — ICAPS 2024_

### Official Review · Reviewer_HGCM · 2024-01-21

**Significance And Importance:** 2
**Soundness:** 4
**Novelty:** 2
**Clarity:** 4
**Overall Evaluation:** 1
**Confidence:** 3

**Weaknesses:**

0: Minor weaknesses requiring some work to be addressed for the paper to be accepted.

**Contributions Of The Paper:**

The paper describes an heuristic search approach for the weight-constrained shortest path problem (WCSP). The approach combines two existing techniques, WC-A* and A*pex, into the WC-A*pex method. The method allows to specify some epsilon threshold and then computes a solution to WCSP that respects the suboptimality bound induced by epsilon. The method itself and additional engineering concepts are described in detail. The experimental evaluation certifies that the new method outperforms previous approaches and studies the sensitivity of the method to the choice of epsilon.

**Ethical Considerations:**

(1) Not Applicable: The paper does not have any ethical considerations to address

**Nomination For Best Paper:**

No

**Questions For Authors:**

Q1. Could you motivate the usage of a doubly linked list to keep track of the elements in open over a data structure with faster access times for individual elements, as e.g. a skip list? For merge operations, the design choice makes a lot of sense but for individual updates, iteration might still be costly.

**Reproducibility:**

4: Authors promise to release code and domains (whichever apply).

**Strengths Of The Paper:**

S1. The studied problem is of high practical relevance and the proposed method for its solution is sensible.

S2. The baselines and the combined method are described in very detail, including pseudo-code and examples.

S3. It is said that the code will be released upon acceptance.

S4. The experimental evaluation shows a significant speed-up over the baselines.

**Weaknesses Of The Paper:**

W1. The proposed methodology is quite incremental and the applied algorithmic and proof concepts are very similar to the ones used for WC-A* and A*pex.

W2. The experimental study only considers two very closely related metrics, namely distance and travel time. Thus, good approximation results are rather unsurprising. In the introduction, examples as electric vehicle and bicycle route planning were mentioned, where energy consumption and height difference play an important role. It would be more interesting and expressive to also consider such metrics in the evaluation. The motivation for minimizing travel time under a distance constraint  is rather weak in my opinion. Note that these two metrics were given in the DIMACS instances not for the purpose of bi-objective path search but to study the performance differences of shortest path planning techniques when using travel time or distance.

---

> ### Author Rebuttal · Authors · 2024-01-28
>
> We thank the reviewer for the valuable feedback and will use them to improve our paper.
>
> We use the DIMACS benchmark because it is used in almost all existing BOSP and WCSP works and is a good starting point. We agree that these instances share a specific "structure" (the specific topology of the graph and the cost are correlated). In fact, a recent survey [1] identified the need for the community to generate a more diverse, publicly available set of benchmarks. But, unfortunately, this is not yet available.
>
> A doubly-linked list makes iterating over Open[s] and removing an element from Open[s] efficient. Currently, in WC-A\*pex, we do not need to access a node based on specific values (such as f-values of apex-path pairs), which we think skip lists are suitable for. However, these are interesting directions for future work. For example, we might only need to iterate over some of the Open[s] based on the g-value of the given apex-path pair and hence save time.
>
> [1] Salzman, Oren, et al. "Heuristic-search approaches for the multi-objective shortest-path problem: progress and research opportunities." IJCAI. 2023.

---

### Official Review · Reviewer_ecKD · 2024-01-21

**Significance And Importance:** 2
**Soundness:** 3
**Novelty:** 3
**Clarity:** 4
**Overall Evaluation:** 2
**Confidence:** 3

**Weaknesses:**

1: Minor weaknesses that are easily fixable.

**Contributions Of The Paper:**

The paper develops bounded suboptimal algorithms for the Weight-Constrained Shortest Path Problem. This includes a straightforward extension of WC-A* to the bounded suboptimal realm, essentially by using it in an anytime fashion and terminating when the incumbent solution provably satisfies the required bound. Second, the A*pex algorithm is modified to work under this setting as well. The correctness of this new algorithm, called WC-A*pex, is proven. Experiments on a set of road networks demonstrate the value of bounded suboptimal search on the problem, as even WC-A* yields meaningful speedups. However, WC-A*pex sees an order of magnitude of additional speedups. Problem-by-problem comparison demonstrate that WC-A*pex is less effective on easy problems than WC-A*, but sees dramatic improvements on the harder problems.

**Ethical Considerations:**

(1) Not Applicable: The paper does not have any ethical considerations to address

**Nomination For Best Paper:**

No

**Questions For Authors:**

1) Any comments on whether bounded suboptimal WC-A*pex is faster or slower than optimal WC-A* on the easy problems?


Post-Response: Thank you for your response. I think including those clarifications would be useful in future iterations of the work.

**Reproducibility:**

4: Authors promise to release code and domains (whichever apply).

**Strengths Of The Paper:**

The paper makes a interesting step forward by bringing bounded suboptimality. WC-A*pex requires a non-trivial, but not too complicated adjustment from the original algorithm. It is generally well explained. The new method is grounded with the necessary theoretical results.

The experimental results are compelling. They do well to show the overall utility of bounded suboptimal search in this domain, and on a reasonably sized test set. I also appreciate the problem-by-problem comparisons which help identify exactly where the improvements are being made.

**Weaknesses Of The Paper:**

I don't have any major weaknesses. The significance of the new problem/method is not the largest perhaps, but I still think it is enough.

I did find the way Lemma 1 and Lemma 2 are described and proven somewhat difficult to understand. I am fairly convinced they are correct, but I struggled to understand parts of them. In particular, I felt they could be more explicit about how the proof was working.

Part of the issue with Lemma 2 was the backwards induction was a bit hard to follow. The approach makes sense in hindsight, but I think it could be clarified more upfront. First, I think the Lemma statement would be clearer as

"For any number of expansions n, if at the beginning of the iteration there exists an expanded AP such that s(AP) = s*_j for some j and ..., then there exists a AP' \in Open such that for some k > j ..."

It's an equivalent statement to what is there, but I think it more clearly identifies the assumptions and implication. I don't think that phrasing I gave is perfect either (the first part still feels like it could be improved), but I feel it helps understand better the rest of the proof.

Next, I found the inductive step hard to understand. I think, the proof needs to be clearer that it is considering the iteration right after AP was expanded, and that once it is true after that step, it remains true for the rest of the search. That last point could also be better stated in the base case.

For the Proof of Lemma 1, it would also help if it was clearer that the proof was by contradiction earlier on. The part starting with "Because AP_sol is not (\epsilon, 0) bounded ... " was not obvious until I got to the end of the proof and realized that the proof was by contradiction. Perhaps stating "The proof is by contradiction. Assume \pi_sol is not (\epsilon, 0) bounded" and then proceeding to derive the contradiction would help with readability.

Beyond that, I had no major issues. Perhaps ideally the methods would be tested on a structurally different problem (ie. not road networks), but what is there is sufficient. Similarly, it would be nice to have an extra image like Figure 4 comparing WC-A*pex with some epsilon >= 0.01 (I don't care which), vs WC-A* with epsilon=0, just to clarify if the main benefit of bounded suboptimality is that it makes the biggest improvements on the hardest problems. I would be surprised otherwise though. More importantly, it would be interesting to see if bounded suboptimal WC-A*pex is faster or slower than optimal WC-A* on the easy problems.


Minor Concerns
- I don't understand why in the base case of Lemma 2, g_2^min(s_goal) is infinity. I understand that s(AP) \neq goal as stated on the next line since we already said that s(AP) = s_{\ell - 1}. But s_goal will have been generated tat this point, so it should be non-infinite. I think this is related to the point above that throughout this proof, it is easy to get confused about if we are at the point where AP was just expanded or not.
- I think Table 1 is averaged over all deltas, which seems a bit odd, but is not a major issue. However, if this is the case, it should be mentioned.

---

> ### Author Rebuttal · Authors · 2024-01-28
>
> We thank the reviewer for the valuable feedback and will improve our paper accordingly.
>
> We will rephrase the lemmas to make them more clearer. We will also make it clear that the proof for Lemma 1 is done through contradiction in the beginning. We will also include a figure that compares WC-A\*pex and WC-A\* in the final version. In the final version, we will clarify that an apex-path pair is considered to be expanded when Alg. 1 reaches Line 8 with it. We will go through the paper to make sure that this is consistently defined.
>
> Regarding minor concerns, the g2min records the minimum g2 values for all expanded nodes, not generated nodes. Therefore, g\_2^min(s\_goal) is infinity before WC-A\*pex expands any apex-path pair on s_goal. On Lines 8-9, Alg 1, we can see that WC-A\*pex will return when it expands the first apex-pair on s_goal.
>
> Table 1 is indeed averaged over all $\delta$s; we will clarify this in the final version. We do so because, from Figure 4, we found that $\delta$ seems not to affect the trend of runtime comparison.
>
> To answer your question, after the submission deadline, we found out that the slowdown was caused by implementation details, where our reported runtime for WC-A\*pex includes the initialization time of some data structures while the reported runtime for WC-A\*(-eps) does not. After fixing this inconsistency, the average/median/variance of speedup factors of WC-A\*pex over WC-A\* on all instances that took WC-A\* less than 0.1 second goes from 0.75/0.52/0.48 to 1.29/1.07/0.76. In other words, WC-A\*pex is slightly faster than WC-A\* on easy problem instances according to our most recent results. We will include the updated results in the final version.

---

### Official Review · Reviewer_xZxu · 2024-01-23

**Significance And Importance:** 1
**Soundness:** 3
**Novelty:** 2
**Clarity:** 3
**Confidence:** 3

**Weaknesses:**

-1: Major weaknesses requiring significant work to be addressed for the paper to be accepted.

**Contributions Of The Paper:**

The paper studies the weight-constrained shortest path problem. In this setting, the problem has two objective functions, and the goal is to minimise one objective while keeping the other under some fixed value. This problem has also been studied in the bi-objective case, and this paper leverages ideas from the bi-objective literature for the constrained case. My impression is that the main contribution is an adaptation of an existing method to this constrained case.

**Ethical Considerations:**

(1) Not Applicable: The paper does not have any ethical considerations to address

**Nomination For Best Paper:**

No

**Overall Evaluation:**

-1: (weak reject)

**Questions For Authors:**

Can you please address my concerns about the novelty, experiments, and writing (see above)?

Furthermore, can you comment on whether you would release the source code?

**Reproducibility:**

1: Difficult to reproduce because of missing detail.

**Strengths Of The Paper:**

Reasonably well motivated problem

The paper provides a step towards heuristic search for the weight-constrained problem. Other works are either optimal methods or generate the full (approximated) pareto front.

The experimental results indicate promising results, however I also have some questions about this (see weaknesses and questions)

The structure of the paper is nice and most parts read well, see further for the concerns about writing.

Overall the approach makes sense

**Weaknesses Of The Paper:**

Novelty

If I understood correct, the main contribution is, briefly, an algorithmic idea where the algorithm merges two states if they are "similar enough". This makes sense. However it seems that previous work (A*pex) used the same idea in the bi-objective case. So this leaves me wondering whether the contribution here is truly new, or an adaptation of an existing and very related method.

The authors also claim a novelty on the data structures used. But as above, it seems that very similar data structures have appeared for the bi-objective problem as well.

Experiments

The paper claims orders of magnitude improvements, which sets very high expectations from the reader side. The experiments indicate promising results, but I feel the experiments do not match the expectations.

The numbers in Table 1 seem impressive at face value. But then the runtime of the competing method (WC-A*) can something be very low (subseconds) on some families of problems. Are these reasonable comparisons? For example, if the avg runtime is 0.07 of the competitor, and yours is 0.02, it is technically a three times speed up, but this is such a small runtime that is hard to see the value. Nevertheless there are cases with longer runtimes, and here the method seems to do well, and does so more consistently.

Is average runtime appropriate? This hides the variance, so it is hard to see the advantage. Perhaps a geometric mean could be useful in this case.

The benchmarks are not motivated, the authors only point out that other papers use them. But are these good benchmarks for comparison? The impact of the threshold on the weight plays an important role here, so I also wonder if this can be chosen in some good and principled way.

There are some impressive runtime improvements, but if I am reading Figure 4 correctly, it also seems that there are cases where there are rather significant slowdowns. It is also not easy to fully process Figure 4, the green cloud of points obstruct seeing the other two parameters. Perhaps it would be better to have one smaller picture for each parameter setting (delta).

"Our preliminary results showed that these improved data structures " -> this seems like a natural point to experiment with properly so that we are sure about the benefits over relying on a preliminary study.

Writing

Overall the paper reads well. I am nevertheless struggling to see the big difference between the bi-objective version and this constrained version. The difference is that the second objective is now given as a constraint rather than the objective, correct? If this is the case, then I would think that it would be easier if the paper presents the work in that way, rather than introducing two different problems. This could help lower the mental load required from the readers, and also saves space.

Minor:

a "different technique" to speed up the search is mentioned in the introduction. It is not clear if this is referring to something new or the previous text.

"dominated by at least one blue dot" -> there is only one blue dot on the figure. I could see that maybe there are two dots that have a blue circle around, it needs to be clarified if these count as blue dots.

---

> ### Author Rebuttal · Authors · 2024-01-28
>
> We thank the reviewer for the valuable feedback and will improve our paper accordingly.
>
> ## Novelty
>
> we want to emphasize again that no previous work has drawn a relation between approximate BOSP algorithms and bounded suboptimal WCSP algorithms, and very few works exist on bounded suboptimal WCSP algorithms. When computing a bounded-suboptimal solution that optimizes a single objective, existing algorithms like WA* or focal search adopt a different technical direction from the one we proposed. Hence, the novelty of this paper lies in tackling a well-motivated problem using a technique that has not been considered. Technically, it is also non-trivial to adopt the merging technique while ensuring that the algorithm is still complete.
>
> ## Experiments
> In Table 1, WC-Apex\* also achieved up to an order of magnitude speedup over the baseline in terms of the max runtime and, in Fig 4, WC-A\*pex outperforms WC-A\*-eps on all instances that took more than 1 second for WC-A\*-eps to solve. In Fig 4, WC-A\*pex shows slowdowns only on easy instances that mostly took both algorithms <0.1 seconds to solve. After the submission deadline, we found out that our reported runtime for WC-A\*pex includes the initialization time of some data structures while the reported runtime for WC-A\*(-eps) does not. After fixing this inconsistency, the average/median/variance of speedup factors of WC-A\*pex over WC-A\*-eps on all instances that took WC-A\*-eps less than 0.1 second goes from 0.64/0.42/0.38 to 1.05/0.88/0.52. We will include the updated results in the final version.
>
> We use the DIMACS benchmark because it is used in almost all existing BOSP and WCSP works. We agree that these instances share a specific “structure” (the specific topology of the graph and the cost are correlated). In fact, a recent survey [1] identified the need for a more diverse, publicly available set of benchmarks. But, unfortunately, this is not yet available.
>
> We will fix the reliability issue of Fig 4 as suggested by the reviewer and add experimental results for the vanilla WC-A\*pex which does not implement the improvements.
>
> ## Writing
> We thank the reviewer for the suggestions and issues pointed out. We will modify the final version accordingly.
>
> ## Reproducibility
> We will release the source code and all the problem instances we used.
>
> [1] Salzman, Oren, et al. "Heuristic-search approaches for the multi-objective shortest-path problem: progress and research opportunities." IJCAI. 2023.

---

### Meta-Review · Area_Chair_eTQg · 2024-02-05

**Recommendation:** Accept (Poster)
**Confidence:** 2

**Metareview:**

The paper combines two existing algorithms to solve the problems at hand. The reviewers agree that the contribution is interesting and the experimental performance shows the value of the approach, but opinions diverge on whether the contribution is sufficiently novel to warrant publication. All reviewers agree that the novelty of the contribution is limited, with one reviewer arguing that the combination is fairly straightforward, the problem at hand is very similar to the bi-objective case and the experimental evaluation oversells the results by mentioning orders of magnitude of difference while the runtimes are in general very short to begin with (for example 0.07s vs 0.02s). On the other hand, one reviewer pointed out that the combination required a nontrivial strategy on when to merge states and that they understood the order of magnitude claim to rather be relative to optimal algorithms, and one reviewer highlighted the high relevance of the topic for many practical applications.

In my opinion the positive aspects of the paper slightly outweigh the negative aspects and I thus recommend acceptance, but this is a weak recommendation since the limited novely is a noticeable concern and I agree that it is in general hard to glean too much from experiments where the runtime is usually less than a second, but this may also just be an unfortunate limitation of the available benchmarks. In case of rejection I support the suggestions of one of the reviewers that the authors could turn the paper into a short paper which would be a better fit for the size of the contribution.

Regarding poster vs oral presentation, I rather suggest a poster presentation since I could imagine that it is easier to show how the algorithm works on a poster where much information can be displayed at once, but this is also a weak recommendation.

**Ethical Considerations:**

(1) Not Applicable: The paper does not have any ethical considerations to address